# Personal Tokens Matter: Towards Token-Aware Training for Personalized LLMs

## Abstract

With large language models (LLMs) now performing strongly across diverse tasks, there is growing demand for them to personalize outputs for individual users. Personalization is typically framed as an additional layer on top of a base NLP task, requiring models to meet user-specific needs while still completing the underlying task. This overlay nature means that many tokens in a response serve the base task, while only a subset encode user-specific information. We term these *personal tokens*, as they are essential for rendering responses personalized. However, their varying positions and contents across scenarios make them difficult to detect directly. To address this challenge, we propose **PerContrast**, a causal intervention–based method that identifies personal tokens by measuring each output token's dependence on user-specific information, achieving up to 87.8% F1 score on our benchmark. Building on this insight, we develop the **PerCE** loss, which adaptively upweights personal tokens during training through an bootstrap procedure, enabling the model to alternately identify and optimize these tokens. Experiments on multiple LLMs show that PerCE substantially improves personalization performance with minimal additional cost, yielding average gains over 10% and up to 40.57% on the LongLaMP dataset, along with strong cross-task and cross-scenario transferability. These results highlight the critical role of personal tokens and establish token-aware training as a simple yet effective paradigm for advancing personalized LLMs.

## 1 Introduction

Large language models (LLMs) have demonstrated remarkable capabilities across a wide range of tasks, such as dialogue (Bubeck et al., 2023; Ding et al., 2023), question answering (Trivedi et al., 2022; Salemi & Zamani, 2025), and logic reasoning (Guo et al., 2025; Trung et al., 2024). This progress has led to widespread deployment of LLMs in user-facing applications. As a result, there is a growing demand for LLMs to not only excel at general tasks, but also to *personalize* their outputs to individual users. In other words, beyond producing generally correct answers, users increasingly expect LLMs to tailor responses according to the user's profile, preferences, or interaction histories (Zhang et al., 2024; Xu et al., 2025b).

Many works have been devoted to this goal. A range of efforts focus on synthesizing personalized data to train LLMs across diverse scenarios, such as personalized text generation (Salemi et al., 2024b; Kumar et al., 2024), personalized question answering (Salemi & Zamani, 2025), and personalized dialogue (Wu et al., 2024). Numerous studies enhance personalization by retrieving and embedding user-specific information to ground model outputs in context (Au et al., 2025; Salemi et al., 2024a; Bu et al., 2025; Zhang et al., 2025; Li et al., 2024; Richardson et al., 2023; Pan et al., 2025). Collectively, these efforts have advanced the development of personalized LLMs. However, a distinctive characteristic of personalization tasks has so far been largely overlooked.

Unlike other tasks, personalization is typically framed as an additional layer on top of a base NLP task, requiring models to address user-specific needs while still accomplishing the underlying task (Salemi et al., 2024b; Kumar et al., 2024; Zhao et al., 2025; Xu et al., 2025a). This overlay nature means that many tokens in the response serve the base task, while *only a subset contribute to personalization*; we refer to these as *personal tokens*. As exemplified in Figure 1, in personalized abstract generation or topic writing, personal tokens mainly capture a user's stylistic preferences, whereas

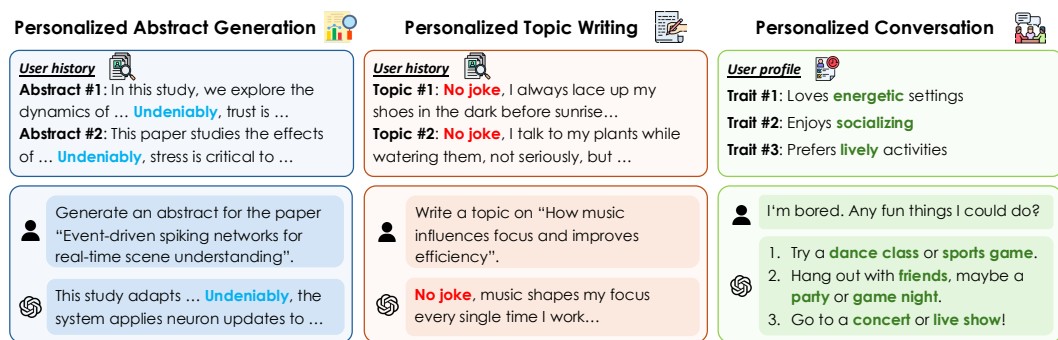

Figure 1: **Examples of personal tokens in personalized tasks.** Personal tokens take diverse forms and may appear in different parts of the response, reflecting user-specific information such as expression style, preferences, or personal attributes.

in personalized conversation, they reflect individual traits. These tokens are the key elements that render a response personalized. Therefore, it is natural to place greater emphasis on personal tokens when enhancing personalized LLMs. However, existing training practices typically treat all tokens uniformly (Kumar et al., 2024; Salemi & Zamani, 2025), which may dilute the emphasis on personalization.

In this work, we address this gap by explicitly identifying and re-weighting personal tokens to improve training. First, we introduce **PerContrast**, a principled method for personal token identification. The core idea is to measure how much each token in the task response depends on user-specific information in the prompt via causal intervention. Concretely, for a given response token, we compare the model's predicted probability of that token when conditioned on the full personalized instruction versus a modified instruction with the personal information removed. If the likelihood drops significantly without the persona, the token is deemed a personal token. This token-level causal analysis enables us to pinpoint which parts of the output are driven by personalization. To evaluate this, we construct a benchmark for token-level personalization and show that PerContrast identifies personal tokens with a high F1 score (up to 87.8%).

We believe that the mechanism for identifying personal tokens can greatly benefit training for personalization. To this end, we propose the **PerCE** loss, which upweights personal tokens that are estimated by the model itself. In this way, PerCE bootstraps personalization by alternating between estimating and optimizing personal tokens. Experimental results across multiple LLMs show that PerCE consistently outperforms the conventional Cross Entropy (CE) loss with minimal additional cost. It achieves a maximum gain of 40.57% and an average improvement of over 10% across all models on LongLaMP dataset. Moreover, PerCE demonstrates significantly stronger transferability in personalization, achieving gains of up to 50% in many cross-task and cross-scenario settings.

In conclusion, the main contributions of this paper are:

- We introduce the concept of personal tokens and highlight their widespread presence in personalization tasks. Furthermore, we propose **PerContrast**, a efficient self-contrast method for personal token identification that achieves up to 87.8% F1 score on our benchmark.

- We develop the **PerCE** loss, which enhances the model's personalization capabilities in an expectation–maximization (EM) manner. PerCE allows the model to alternately perform online self-contrast weighting and optimization at each training step, thereby encouraging it to automatically place greater focus on personalized information.

- We conduct extensive experiments across multiple models with scale from 4B to 14B on a wide range of settings. The results show that PerCE substantially improves the performance and generalization in personalization scenarios while incurring minimal additional cost.

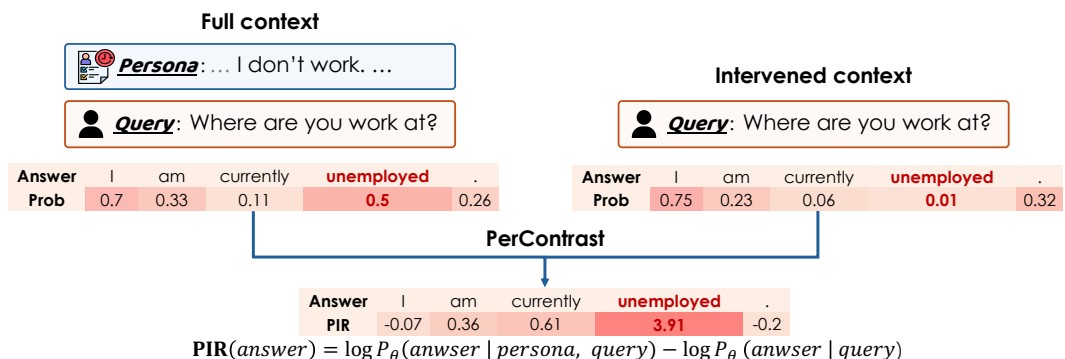

$$\text{PIR}(answer) = \log P_\theta(anwser \mid persona, \ query) - \log P_\theta(anwser \mid query)$$

Figure 2: Illustration of PerContrast. PIR stands for Personal Influence Ratio (Equation 2), and Red words refers to identified personal tokens.

## 2 PERCONTRAST: PRECISE PERSONAL TOKEN IDENTIFICATION

In this section, we introduce **PerContrast**, a self-contrast method for effective and efficient identification of personal tokens. We begin with the main design of PerContrast. Next, we construct a synthetic experiment to verify the reliability of PerContrast on identifying personal tokens.

### 2.1 EXTRACTING PERSONAL TOKENS VIA SELF-CONTRAST

LLM personalization aims to adapt generic models to user-specific preferences while maintaining strong performance on the base task. Unlike standard NLP tasks, where outputs are conditioned only on the input query, personalization additionally depends on the user persona $p_u$. Formally, given an input query $x$ and a user persona $p_u$, a personalized LLM models the conditional probability distribution:

$$P_\theta(y \mid p_u, x), \tag{1}$$

where $y$ is the generated response and $\theta$ are the model parameters. For evaluation, the response is either compared with a reference response $y_r$ or assessed directly by an LLM judge.

As illustrated in Figure 1, within either $y_r$ or $y$, only a subset of tokens, referred to as *personal tokens*, directly encode user-specific preferences. These tokens vary in both position and content across personalization tasks, which makes them difficult to detect directly. To address this challenge, we propose PerContrast, a principled method for accurately identifying personal tokens, which in turn provides a foundation for emphasizing them in the training of personalized LLMs.

This motivates us to find a surrogate metric that can accurately identify the personal tokens. We tackle this challenge with our proposed method PerContrast, which enables accurate identification of personal tokens and lays the foundation for emphasizing them in personalized LLMs.

**Personalization Intervention.** To quantify the influence of a user persona on each response token $y_i$, we perform an *intervention* on the model context. Specifically, given a personalization task with query–response pair $(x, y)$ and a language model $P_\theta$ (with strong personalization ability), we compute the difference between the log probability of $y_i$ with the full persona $p_u$ and the log probability without the persona:

$$\text{PIR}(y_i; \theta) = \log P_\theta(y_i \mid p_u, x, y_{<i}) - \log P_\theta(y_i \mid x, y_{<i}). \tag{2}$$

We refer to this measure as the **Personal Influence Ratio (PIR)**, which captures the contribution of the user persona to the prediction of each token.

From a causal perspective, the second term in Equation 2 serves as the counterfactual outcome obtained by the intervention (removing the user persona). Accordingly, PIR estimates the individual treatment effect (ITE) (Pearl, 2010) of the persona under the language model $P_\theta$. A high PIR value indicates that the persona plays an important role in predicting $y_i$, thereby identifying it as a key token in personalization tasks, as shown in Figure 2.

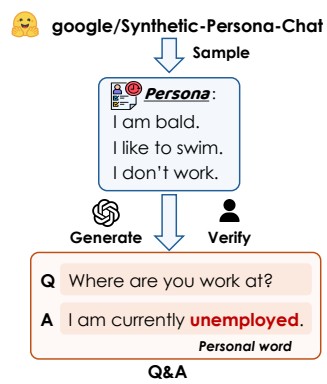

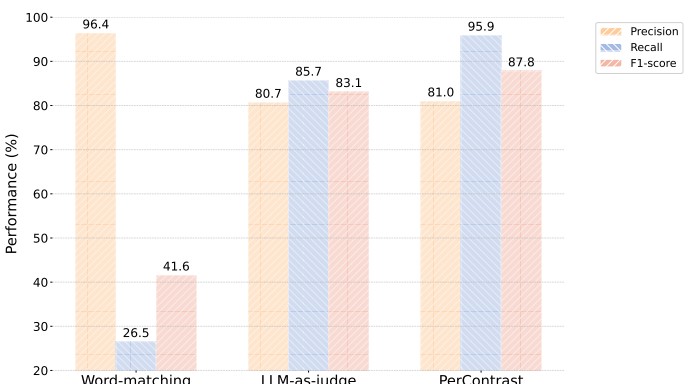

Figure 3: Pipeline of dataset construction.

Figure 4: Performance comparison across three methods on personal tokens identification.

## 2.2 PerContrast Achieves Superior Performance in Identifying Personal Tokens

After introducing PerContrast, we want to quantitatively verify its ability to identify personal tokens. As the concept of personal tokens was first introduced by us, there is currently no annotated benchmark to evaluate the identification accuracy of personal tokens. Therefore, we construct a synthetic personalized Q&A dataset and conduct evaluation on it. Experimental results demonstrate that PerContrast outperforms two intuitive baselines: a word-matching method and an LLM-as-judge approach. Implementation details are provided in Appendix B.2.

**Synthetic Dataset.** We construct a synthetic dataset as illustrated in Figure 3. Specifically, we sample 100 personas from the Synthetic-Persona-Chat dataset (Jandaghi et al., 2024), which contains many synthetic user personas. For each persona, we prompt GPT-4o to generate a question and then produce an answer of 5–20 words. This answer can be regarded as a simple personalized response, which naturally contains some personal tokens. We simultaneously ask GPT-4o to identify the personal tokens from the answers as the ground-truth of personal tokens. These tokens are then verified by human annotators to ensure accuracy. Through this process, we obtain 100 question–answer pairs with about 200 carefully annotated ground-truth personal tokens.

**Baselines.** For comparison, we introduce two intuitive baselines, namely word-matching method and LLM-as-judge approach. The word-matching method treats all content words (e.g., nouns, verbs, adjectives, etc.) that appear in both the persona and the answer as personal tokens. LLM-as-judge refers to letting the same LLM with PerContrast determine the personal tokens through generation, given the persona and the Q&A pair.

**Results.** As shown in Figure 4, we report the precision, recall, and F1-score of all three methods. First, although the word-matching method achieves a high precision of 96.4%, it struggles to identify all personal tokens because personalized content does not always appear in the form of exact word matches. Consequently, it suffers from a low recall of 26.5%, resulting in a low F1-score of 41.6%. In comparison, both the LLM-as-judge approach and PerContrast achieved satisfactory results. However, while the two methods demonstrate comparable precision (**81.0%** *vs.* 80.7%), PerContrast attains a higher recall (**95.9%** *vs.* 85.7%), indicating a more comprehensive identification of personal tokens, which leads to a superior F1-score (**87.8%** *vs.* 83.1%). Furthermore, since LLM-as-judge requires additional model generation whereas PerContrast necessitates only one extra forward pass, PerContrast is significantly more efficient. In our experiment, PerContrast is about **10 times faster** than the LLM-as-judge method. In summary, PerContrast demonstrates superior performance and efficiency in identifying personal tokens over existing methods.

## 3 PerCE: Enhancing Personalized LLMs with Personal Tokens

By default, Cross-Entropy (CE) loss remains the predominant objective for training personalized LLMs (Salemi et al., 2024b; Tan et al., 2024). Given an input query $x$, a user persona $p_u$, and the

reference response $y_r$, the CE loss takes a uniform average over all tokens:

$$\text{CE}(y; \theta) = -\frac{1}{n} \sum_{i=1}^{n} \log P_\theta(y_i \mid x, y_{<i}). \tag{3}$$

This objective encourages the model to reproduce reference outputs that capture both task requirements and user-specific preferences. However, CE treats all tokens equally, while personal tokens often constitute only a small fraction of the output. Consequently, these tokens are likely to be insufficiently optimized under this loss, which limits improvements in personalization performance and generalization.

Following the design of PerContrast, we introduce **PerCE**, which performs ***online token re-weighting*** in a ***bootstrap*** manner. Specifically, the ***trained model itself*** computes token weights during each training step, thereby dynamically assessing the importance of each token on the fly. Formally, PerCE re-weights each token $y_i$ according to its contribution $w(y_i; \theta)$ to personalization:

$$\text{PerCE}(y; \theta) = -\frac{1}{n} \sum_{i=1}^{n} w(y_i; \theta) \log P_\theta(y_i \mid p_u, x, y_{<i}), \tag{4}$$

where $w(y_i; \theta)$ is a token-level influence function $w : \mathbb{X} \to [m, M]$, defined by the likelihood ratio between the predicted probability with and without the user persona (i.e., $\text{PIR}(y_i; \theta)$ in Section 2.1):

$$w(y_i; \theta) = \text{clip}\Big(\text{PIR}(y_i; \theta),\ m, M\Big)$$

$$= \text{clip}\Bigg(\log P_\theta(y_i \mid p_u, x, y_{<i}) - \log P_\theta(y_i \mid x, y_{<i}),\ m, M\Bigg). \tag{5}$$

The bounds $[m, M]$ are predefined hyperparameters that prevent numerical instability during training.

With $\text{PIR}(y_i; \theta)$ measuring the impact of user persona $p_u$ on predicting token $y_i$, those persona-dependent tokens will be assigned larger weights, proportionally to the personal informativeness. By this, ***PerCE bootstraps the model's personalization capabilities in an EM (expectation-maximization) way***: the language model $P_\theta$ first uses itself to estimate persona influence of each token $w(y_i; \theta)$ (Equation 5); and we then use this estimate to update the model parameters by optimizing the PerCE.

## 4 EXPERIMENTS

To evaluate PerCE, we compare its performance with standard CE loss on the LongLaMP dataset across multiple LLMs. The results show that PerCE achieves significant improvements, with a maximum gain of 40.57% and an average improvement of over 10% across all models. Moreover, PerCE demonstrates strong generalization, achieving gains of up to 50% in many cross-task and cross-scenario settings. Importantly, these substantial gains come at minimal cost, since PerCE introduces only a slight overhead by requiring just one additional forward pass with a short persona-removed.

**Dataset.** To assess the effectiveness of our approach, we use the LongLaMP dataset (Kumar et al., 2024), which focuses on personalized text generation. The dataset contains three tasks: Personalized Abstract Generation (PAG), Personalized Review Writing (PRW), and Personalized Topic Writing (PTW).[1] Considering the large scale of the dataset, we randomly sample 2,000 training examples and 500 test examples for each task. To verify the transferability of our method, we further evaluate on the ALOE Benchmark (Wu et al., 2024), which contains 100 carefully curated multi-turn conversations with distinct user personas. Unlike LongLaMP, where user persona is explicitly included in the prompt, ALOE does not provide user information directly, requiring models to infer user preferences implicitly through multi-turn dialogue.

---

[1]The dataset also includes a personalized email completion task, which relies on private data and is therefore excluded from our experiments.

Table 1: Performance (ROUGE-L) of Qwen3-4B, Qwen3-14B and Llama3-8B fine-tuned with standard CE and PerCE on personalized generation tasks from LongLaMP, including Abstract Generation, Review Writing, and Topic Writing. Each model is trained separately on its corresponding task.

| Model | Method | Personalized Abstract Generation | Personalized Review Writing | Personalized Topic Writing | Average |
|---|---|---|---|---|---|
| GPT-4o | — | 0.2904 | 0.2099 | 0.1924 | 0.2309 |
| GPT-4o-mini | — | 0.2771 | 0.2088 | 0.1882 | 0.2247 |
| Qwen3-4B | CE | **0.3727** | 0.1898 | 0.1665 | 0.2430 |
| | PerCE | 0.3619 | **0.2668** | **0.2102** | **0.2796** |
| | **Gain** | -2.90% | +40.57% | +26.25% | +15.08% |
| Qwen3-14B | CE | **0.3865** | 0.2321 | 0.1805 | 0.2664 |
| | PerCE | 0.3785 | **0.2784** | **0.2307** | **0.2959** |
| | **Gain** | -2.07% | +19.95% | +27.81% | +11.07% |
| Llama3-8B | CE | 0.3573 | 0.2311 | 0.2021 | 0.2635 |
| | PerCE | **0.3756** | **0.2771** | **0.2218** | **0.2915** |
| | **Gain** | +5.12% | +19.90% | +9.75% | +10.63% |

**Metric.** For both datasets, we adopt the official evaluation metrics. For LongLaMP, we follow the official protocol and report ROUGE-L to measure the quality of personalized text generation. For ALOE, evaluation is conducted with given LLM-as-a-Judge framework, where an LLM evaluator assesses the degree of alignment to user-specific preferences in multi-turn dialogue on a 1–5 scale.

**Setup.** We conduct experiments on Qwen3-4B, Qwen3-14B (Yang et al., 2025) and Llama3.1-8B-Instruct (Llama3-8B) (Dubey et al., 2024). All models are trained with both standard CE and our PerCE on each task of the LongLaMP dataset. In addition to evaluating on the corresponding task, we test each model on other tasks within LongLaMP to assess cross-task personalization transfer. Furthermore, we evaluate generalization in a distinct multi-turn personalized dialogue scenario to examine the model's cross-scenario personalization capability. Detailed hyperparameters and prompt templates are provided in Appendix B.1 and D.

**Main Result.** As shown in Table 1, we report the performance of 3 models fine-tuned with standard CE and PerCE on LongLaMP, with GPT-4o and GPT-4o-mini included as reference baselines. The results show that by using PerCE, we achieve an average gain of over 10% across the three tasks on all models. In particular, PerCE consistently outperforms standard CE on Review Writing and Topic Writing, with improvements of up to +40.57% and +26.25% on Qwen3-4B. The gains on Abstract Generation are relatively less pronounced, likely because abstract generation is more constrained than open-ended writing tasks and thus offers fewer opportunities for personalization. Nevertheless, the subsequent hyperparameter robustness analysis (Table 2) shows that by emphasizing personal tokens, PerCE also improves training stability. Overall, these results highlight that PerCE effectively enhances personalized text generation across different models and tasks.

**Robustness to Learning Rates.** As shown in Table 2, PerCE consistently delivers stable and strong performance across all tasks under varying learning rates. In contrast, while CE performs reasonably well on the PAG task, it is highly sensitive to learning rate changes: its score drops sharply from 0.3728 to 0.2261 when the learning rate increases from $5e^{-6}$ to $5e^{-5}$, with variance reaching 44.65, indicating severe instability. These results demonstrate that PerCE not only achieves higher average performance but also provides substantially greater robustness to hyperparameter variation.

**Analysis to Clipping Thresholds.** Table 3 reports the performance of Qwen3-4B on the LongLaMP dataset under different clipping thresholds. For our main experiments, we adopt the configuration of Clip Min = 0.8 and Clip Max = 5.0. This is not the best for any single task, which indicates the potential for stronger performance with fine-grained hyperparameter tuning. Overall, the model shows relatively stable results across different settings, suggesting that PerCE is not

Table 2: Performance across different learning rates for Qwen3-4B on the LongLaMP dataset. Green indicates the best results, while red indicates the worst. Variance values are scaled by $10^{-4}$. Personalized Abstract Generation (PAG), Personalized Review Writing (PRW), and Personalized Topic Writing (PTW) are abbreviated due to space limits.

| LR | Task | | |
|---|---|---|---|
| | PAG | PRW | PTW |
| 2e-6 | 0.3247 | 0.2446 | 0.1778 |
| 5e-6 | 0.3469 | 0.2590 | 0.1883 |
| 8e-6 | 0.3561 | 0.2619 | 0.1998 |
| 2e-5 | 0.3619 | 0.2650 | 0.2098 |
| 5e-5 | 0.3585 | 0.2668 | 0.2102 |
| Mean | 0.3496 | 0.2595 | 0.1972 |
| Variance | 1.800 | 0.6231 | 1.580 |

(a) PerCE

| LR | Task | | |
|---|---|---|---|
| | PAG | PRW | PTW |
| 2e-6 | 0.3560 | 0.1705 | 0.1417 |
| 5e-6 | 0.3728 | 0.1805 | 0.1542 |
| 8e-6 | 0.3687 | 0.1862 | 0.1379 |
| 2e-5 | 0.2339 | 0.1898 | 0.1665 |
| 5e-5 | 0.2261 | 0.1722 | 0.1532 |
| Mean | 0.3115 | 0.1798 | 0.1507 |
| Variance | **44.65** | 0.571 | 1.026 |

(b) CE

Table 3: Performance across different clipping thresholds for Qwen3-4B on the LongLaMP dataset. Rows indicate Clip Min and columns indicate Clip Max. Green denotes the best results, red denotes the worst, and Bold denotes the results in our main experiments.

| Min | Max | | | Min | Max | | | Min | Max | | |
|---|---|---|---|---|---|---|---|---|---|---|---|
| | 2 | 5 | 8 | | 2 | 5 | 8 | | 2 | 5 | 8 |
| 0.2 | 0.3247 | 0.3469 | 0.3605 | 0.2 | 0.2692 | 0.2649 | 0.2681 | 0.2 | 0.2112 | 0.2098 | 0.2091 |
| 0.5 | 0.3601 | 0.3620 | 0.3610 | 0.5 | 0.2655 | 0.2680 | 0.2666 | 0.5 | 0.2096 | 0.2107 | 0.2122 |
| 0.8 | 0.3540 | **0.3619** | 0.3625 | 0.8 | 0.2652 | **0.2668** | 0.2666 | 0.8 | 0.2118 | **0.2102** | 0.2082 |

(a) PAG            (b) PRW            (c) PTW

sensitive to the choice of clipping thresholds and maintains consistent effectiveness across all three tasks.

**Cross-Task Generalization.** Table 4 and Table 8 present the cross-task transfer performance of Qwen3-4B and Llama3-8B fine-tuned with standard CE and PerCE, where models are trained on a single source task and evaluated on other target tasks. The results demonstrate that PerCE exhibits strong generalization: even when trained on one task, it consistently improves performance on other tasks compared to CE. For example, on Qwen3-4B trained on PTW, PerCE achieves gains of +56.62% and +49.90% on PAG and PRW over standard CE. Notably, several out-of-domain scores achieved by PerCE even surpass the in-domain scores of CE on the corresponding tasks, such as 0.2211 for PAG → PRW vs. 0.1898 for PRW, and 0.1955 for PRG → PTW vs. 0.1665 for PTW. These results highlight that PerCE achieves robust and superior cross-task generalization.

**Cross-Scenario Transfer.** Personalization needs are inherently cross-scenario, as users expect models to generate preference-aligned outputs regardless of the context. Therefore, cross-scenario transfer is particularly important. To assess this ability, we evaluate the fine-tuned models on the ALOE benchmark. Unlike LongLaMP, where user history is explicitly included in the prompt, ALOE does not directly provide user information. Instead, it allows models to interact with users in multi-turn dialogue and requires them to infer preferences solely from the conversation in order to deliver user-tailored responses. This creates a substantial scenario gap between the two benchmarks. As shown in Table 5 and Table 9, PerCE consistently outperforms CE across nearly all settings with substantial gains, demonstrating significantly stronger transfer performance in cross-scenario personalization.

Table 4: Cross-task transfer performance of Qwen3-4B fine-tuned with standard CE and PerCE. Rows indicate the source task used for training, while columns represent the target tasks used for evaluation. Detailed results for Llama3-8B are provided in Table 8.

| Source Task | Method | Target Task | | |
|---|---|---|---|---|
| | | PAG | PRW | PTW |
| PAG | CE | **0.3727** | 0.1744 | 0.1617 |
| | PerCE | 0.3619 | **0.2211** | **0.1657** |
| | **Gain** | -2.90% | +26.78% | +2.47% |
| PRW | CE | 0.2055 | 0.1898 | 0.1522 |
| | PerCE | **0.3290** | **0.2668** | **0.1955** |
| | **Gain** | +60.10% | +40.57% | +28.45% |
| PTW | CE | 0.2063 | 0.1443 | 0.1665 |
| | PerCE | **0.3231** | **0.2163** | **0.2102** |
| | **Gain** | +56.62% | +49.9% | +26.25% |

Table 5: Cross-scenario transfer performance of Qwen3-4B models fine-tuned with CE and PerCE. Rows correspond to the source task used for training, and columns indicate the performance of dialogue turns $k$ in the ALOE evaluation. Detailed results for Llama3-8B are provided in Table 9.

| Source Task | Method | ALOE | | | | | |
|---|---|---|---|---|---|---|---|
| | | k=6 | k=7 | k=8 | k=9 | k=10 | Average |
| PAG | CE | **2.55** | 2.52 | 2.35 | 2.33 | 2.40 | 2.43 |
| | PerCE | **2.55** | **2.55** | **2.60** | **2.52** | **2.60** | **2.56** |
| | **Gain** | 0.00 | +0.03 | +0.25 | +0.19 | +0.20 | +0.13 |
| PRW | CE | 1.25 | 1.23 | 1.32 | 1.27 | 1.27 | 1.27 |
| | PerCE | **2.83** | **2.80** | **2.65** | **2.75** | **2.85** | **2.78** |
| | **Gain** | +1.58 | +1.57 | +1.33 | +1.48 | +1.58 | +1.51 |
| PTW | CE | 1.07 | 1.05 | 1.05 | 1.07 | 1.12 | 1.07 |
| | PerCE | **2.80** | **2.90** | **2.70** | **2.67** | **2.73** | **2.76** |
| | **Gain** | +1.73 | +1.85 | +1.65 | +1.60 | +1.61 | +1.69 |

**Training Efficiency.** Although we do not apply any engineering optimizations, [2] the training time of PerCE is only slightly higher than that of the vanilla CE loss, as shown in Table 6. Given the substantial improvements in performance and generalization brought by PerCE, this overhead is entirely acceptable. In detail, PerCE introduces an additional forward pass at each training step, but this pass only processes a short persona-removed context. In most personalization tasks, user persona typically occupies a substantial portion of the context window. For LongLaMP, removing the persona reduces the input length to only about 7%, as shown in Figure 7. Consequently, PerCE incurs minimal computational and time overhead in practice, making it highly practical for real-world applications.

## 5 RELATED WORK

**Diverse Scenarios of LLM Personalization.** Personalization is typically framed as an additional layer on top of standard NLP tasks, where models must not only solve the task but also adapt to user personas (Zhang et al., 2024; Xu et al., 2025b). For example, benchmarks such as Salemi et al. (2024b); Kumar et al. (2024); Zhao et al. (2025); Afzoon et al. (2024); Salemi & Zamani (2025); Wu et al. (2024) construct user personas and tasks across dialogue, text generation, and question answering, and evaluate LLM personalization in these settings. However, most existing works focus on individual scenarios (Salemi et al., 2025; Lee et al., 2024; Magister et al., 2024),

---

[2]We only linearly add the weight-computing step into the standard training pipeline.

Table 6: Per-step training time (s) of Qwen3-4B on 4×A100 GPUs with batch size 3 per GPU and gradient accumulation of 8.

| Method | PAG | PRW | PTW | Average |
|---|---|---|---|---|
| CE | 59.52 | 59.06 | 61.89 | 60.16 |
| PerCE | 67.41 | 71.42 | 69.52 | 69.45 |

Table 7: Prompt length with and without user persona across three personalized tasks, tokenized by Qwen3Tokenizer.

| Setting | PAG | PRW | PTW | Average |
|---|---|---|---|---|
| w. Persona | 975.1 | 2062.6 | 1055.3 | 1364.3 |
| w.o. Persona | **64.5** | **168.9** | **54.8** | **96.1** |

overlooking the shared characteristics of personalization and the potential for cross-task or cross-scenario transferability. We argue that the proposed concept of personal tokens represents an initial step toward addressing this gap.

**Methods for Improving LLM Personalization.** Most efforts to improve LLM personalization focus on retrieving and integrating user information into the model (Au et al., 2025; Salemi et al., 2024a; Bu et al., 2025; Zhang et al., 2025; Li et al., 2024; Richardson et al., 2023; Pan et al., 2025). Another line of work constructs personalized datasets through curated or synthetic pipelines to better train LLMs (Ge et al., 2024; Jandaghi et al., 2024; Wu et al., 2024). A smaller body of research develops training algorithms specifically designed to enhance how LLMs understand and leverage user information (Poddar et al., 2024). The concept of personal tokens introduced in this work represents a fundamental feature of personalization tasks and offers a novel perspective that complements these directions, with the potential to further strengthen LLM personalization.

**Re-Weighting Methods for LLM Training.** Re-weighting methods for LLM training have been extensively studied, primarily with the aims of enhancing model performance (Lin et al., 2024a; Fang et al., 2024; Lin et al., 2024b), improving training efficiency (Clark et al., 2022), and mitigating token imbalance (Luo et al., 2023; Hu et al., 2023; Gu et al., 2020; Wang et al., 2020). A complementary line of research explores re-weighting through data selection, addressing challenges such as data quality (Li et al., 2023a; Iskander et al., 2024), data diversity (Liu et al., 2023), and distribution alignment (Li et al., 2023b; Ni et al., 2024). Despite these advances, no prior work has applied re-weighting specifically to improve LLM personalization. In contrast, our approach directly re-weights tokens according to their dependence on personal information, providing a more principled and targeted solution.

## 6 CONCLUSION AND DISCUSSION

In this work, we emphasize the overlay nature of personalization tasks and highlight the overlooked role of personal tokens—the subset of response tokens that explicitly encode user-specific information. We propose PerContrast, a principled token-level identification method based on causal intervention, and demonstrate its effectiveness in detecting personal tokens with high accuracy. Building on this mechanism, we introduce the PerCE loss, which re-weights training toward these tokens in an expectation–maximization manner. Experiments across diverse models and personalization tasks show that PerCE substantially improves personalization performance and generalization while incurring minimal overhead. These findings underscore the importance of token-aware training for advancing personalized LLMs.

Despite the strong improvements achieved by PerCE, we believe the potential of personal tokens extends far beyond our current exploration. As an essential feature of personalization tasks, personal tokens open a novel perspective for advancing LLM personalization. They could play a role across multiple stages of the personalization pipeline—for instance, serving as fine-grained supervisory signals for learning user embeddings, or providing more informative training objectives for user-specific PEFT methods. Exploring these directions may open new avenues for building more adaptive, robust, and user-aligned language models, and we hope our work can serve as a foundation for this line of research.

## ETHICS STATEMENT

We are not aware of any specific ethical concerns regarding this work. All experiments are conducted on publicly available or synthetic datasets, without the use of sensitive, private, or proprietary information. We have ensured that the methods and findings are presented responsibly.

## REPRODUCIBILITY STATEMENT

We provide complete details of our methods, hyperparameters, datasets, and evaluation metrics in both the main paper and the appendix. To further support transparency and reproducibility, we will release our code upon acceptance.

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

Table 8: Cross-task transfer performance of Llama3-8B fine-tuned with standard CE and PerCE. Rows indicate the source task used for training, while columns represent the target tasks used for evaluation.

| Source Task | Method | Target Task | | |
|---|---|---|---|---|
| | | PAG | PRW | PTW |
| PAG | CE | 0.3573 | 0.1554 | 0.1590 |
| | PerCE | **0.3756** | **0.2036** | **0.1803** |
| | **Gain** | +5.12% | +31.02% | +13.40% |
| PRW | CE | 0.1827 | 0.2311 | 0.1971 |
| | PerCE | **0.3390** | **0.2771** | **0.2032** |
| | **Gain** | +85.55% | +19.90% | +3.09% |
| PTW | CE | 0.2276 | 0.2225 | 0.2021 |
| | PerCE | **0.3368** | **0.2507** | **0.2218** |
| | **Gain** | +47.98% | +12.67% | +9.75% |

Table 9: Cross-scenario transfer performance of Llama3-8B models fine-tuned with CE and PerCE. Rows correspond to the source task used for training, and columns indicate the performance of dialogue turns $k$ in the ALOE evaluation.

| Source Task | Method | ALOE | | | | | |
|---|---|---|---|---|---|---|---|
| | | k=6 | k=7 | k=8 | k=9 | k=10 | Average |
| PAG | CE | 1.82 | 1.95 | 1.90 | 1.98 | 1.82 | 1.89 |
| | PerCE | **2.80** | **2.40** | **2.62** | **2.52** | **2.40** | **2.55** |
| | **Gain** | +0.98 | +0.45 | +0.72 | +0.54 | +0.58 | +0.66 |
| PRW | CE | 1.02 | 1.10 | 1.00 | 1.18 | 1.12 | 1.08 |
| | PerCE | **2.88** | **2.77** | **2.77** | **2.75** | **2.60** | **2.75** |
| | **Gain** | +1.86 | +1.67 | +1.77 | +1.57 | +1.48 | +1.67 |
| PTW | CE | 1.05 | 1.12 | 1.20 | 1.20 | 1.20 | 1.15 |
| | PerCE | **3.08** | **2.95** | **2.98** | **2.90** | **2.98** | **2.98** |
| | **Gain** | +2.03 | +1.83 | +1.78 | +1.70 | +1.78 | +1.83 |

## A  ADDITIONAL EXPERIMENTAL RESULTS

**Transfer Results on Llama3-8B.** In addition to the results on Qwen3-4B (Tables 4 and 5), we also conduct transfer experiments on Llama3-8B to assess the generalizability of our approach across model families. Table 8 presents cross-task transfer performance on the personalized generation tasks from LongLaMP, while Table 9 reports cross-scenario transfer performance on multi-turn conversations from ALOE. Across both settings, PerCE consistently outperforms standard CE on Llama3-8B, demonstrating that our method not only strengthens personalization but also generalizes effectively beyond a single model family.

## B  EXPERIMENTAL DETAILS

### B.1  HYPERPARAMETER SETUP OF EXPERIMENTS IN TABLE 1

During training, we use a batch size of 32 for Qwen3-14B, 64 for Llama3-8B and 96 for Qwen3-4B. We perform a grid search over the learning rate ranges specified in Table 10. During inference, we set the temperature to 0.4 and the maximum number of generated tokens to 512.

Table 10: The hyperparameter list

| Hyperparameter | Value |
|---|---|
| *Training* | |
| Batch Size | [32, 64, 96] |
| Epoch | 3 |
| Optimizer | AdamW |
| Learning Rate | [2e-6, 5e-6, 8e-6, 2e-5, 5e-5] |
| AdamW Betas | (0.9, 0.999) |
| Warmup Ratio | 0.04 |
| Max Length | 5000 |
| Clip Max | 5.0 |
| Clip Min | 0.8 |
| *Inference* | |
| Sampling Temperature | 0.4 |
| Max New Tokens | 512 |

## B.2 SYNTHETIC EXPERIMENT

For PerContrast, we utilize Qwen3-30B-A3B-Instruct-2507 (Yang et al., 2025) to calculate the contrastive probabilities. Tokens with a PIR value exceeding 1 are considered personal tokens. For the word-matching method, we use the Python "nltk" package to extract content words. For LLM-as-judge, we also use Qwen3-30B-A3B-Instruct-2507 to obtain the personal words through the prompt template as shown in Appendix D. For all methods, as long as an identified personal token is contained within a ground-truth personal word, we consider that method to have correctly recognized the personal token.

## C THE USE OF LARGE LANGUAGE MODELS (LLMs)

In this work, LLMs are primarily employed for polishing the language of the manuscript to improve clarity and readability. All conceptual development, experimental design, and result interpretation are conducted independently by the authors. The use of LLMs is strictly limited to auxiliary tasks, ensuring that the scientific contributions of this paper remain entirely unaffected by such tools.

## D  PROMPT TEMPLATE

Here we provide the prompts template used in our experiments.

### D.1  PROMPTS IN LONGLAMP TASKS

---

**Prompt for Personalized Abstract Generation**

**System:**
Your task is to write abstracts for the user's paper. The following are abstracts from the user's previous papers:

{previous_abstracts}

Based on these examples, compose a new abstract that is consistent with the user's preferred style and language.

**User:**
{question}

**Assistant:**

---

**Prompt for Personalized Review Writing**

**System:**
Your task is to write product review texts for the user. The following are the previous reviews from the user:

{previous_reviews}

Based on these examples, compose a new product review that is consistent with the user's preferred style and tone.

**User:**
{question}

**Assistant:**

---

**Prompt for Personalized Topic Writing**

**System:**
Your task is to write on a given topic for the user. The following are examples of the user's previous writings:

{previous_writings}

Based on these examples, write a new text on the given topic that is consistent with the user's preferred style and tone.

**User:**
{question}

**Assistant:**

---

## D.2 PROMPTS FOR SYNTHETIC DATASET

---

**Prompt for the synthetic data construction**

**User:**
Now I have a paragraph of a person's persona:

{persona}

I want you to write a short question to this person and a corresponding short answer. Note that the answer should be concise, able to be inferred from his/her persona (but not the question), and may have words that are different from those in the persona. Highlight the keywords in the answer that reflect his/her persona. The keywords should also be short, not the whole sentence.

**Assistant:**

---

**Prompt for the LLM-as-judge method**

**User:**
This is the persona of user 1:

{persona}

This is the conversation between two users:

User 2: {user 2's speech}
User 1: {user 1's speech}

Identify the keywords in user 1's speech in the conversation that reflect his/her persona (but not words in the description of his/her persona).

Keywords:

**Assistant:**

---

# E  DATASET EXAMPLES

Here we provide several examples of the synthetic dataset in our experiments.

---

**Example of the synthetic dataset**

**Persona:**
I am thin.
I like to hunt.
I love the tigers baseball team.
I am blonde.
I like the tv show the walking dead.

**Question:**
Ever catch anything on your hunts?

**Answer:**
Got a deer last season.

**Ground truth:**
deer

---

**Example of the synthetic dataset**

**Persona:**
I am short.
My hair is brown.
I am not thin.
I like to sew.

**Question:**
What's your favorite way to spend a free afternoon?

**Answer:**
I usually stitch clothes or crafts.

**Ground truth:**
stitch

---

**Example of the synthetic dataset**

**Persona:**
My favorite color is red.
I live with my parents.
I prefer headsets over earbuds.
I travel often.
I have an iphone.

**Question:**
What's your go-to device for listening to music or calls?

**Answer:**
I like the ones that cover my ears.

**Ground truth:**
cover my ears

---

