# OpenReview forum: "Personal Tokens Matter: Towards Token-Aware Training for Personalized LLMs"
_ICLR.cc/2026/Conference — Submitted to ICLR 2026_

### Official Review · Reviewer_eKUJ · 2025-10-26

**Soundness:** 2
**Presentation:** 3
**Contribution:** 2
**Rating:** 4
**Confidence:** 4

**Summary:**

This paper addresses the emerging challenge of improving personalization in LLMs. The authors observe that most personalization approaches treat all output tokens uniformly, even though only a small subset of personal tokens can carry user-specific information. To better capture this distinction, the paper introduces PerContrast and PerCE. The paper reports extensive experiments on the LongLaMP dataset.

**Strengths:**

1. Personalization represents a critical direction for the future of LLMs, and this work tackles a highly relevant question, i.e., how to identify and optimize the specific parts of model outputs that encode personal information.
2. The proposed approach, namely leveraging the difference in token likelihoods with and without persona information, is straightforward, theoretically grounded, and easy to reproduce in practice. The PerContrast and PerCE framework offers a clear and interpretable mechanism for focusing on personalization-relevant tokens.
3. The experiments convincingly demonstrate that emphasizing personal tokens improves model performance on personalized generation tasks, with significant gains in ROUGE-L and better transfer to unseen personalization scenarios.

**Weaknesses:**

1. While the idea of personal tokens is appealing, it remains unclear whether such tokens exist distinctly in natural user data. On synthetic or benchmark datasets with explicitly defined personas, identifying these tokens may be straightforward, but real-world user data often contain subtler, more diffuse personalization cues.

2. My concerns about the experiment:

   - The experimental comparisons are somewhat weak. The paper benchmarks PerCE only against the standard cross-entropy baseline, which is too simple to establish the true effectiveness of the method. Including comparisons with other recent personalization techniques (e.g., retrieval-based, PEFT-based, or preference-alignment methods) would strengthen the claims.

   - Evaluation is limited to the LongLaMP dataset (and a brief transfer to ALOE). Broader validation on other personalization datasets, such as LaMP, would enhance the robustness of the results.

3. A recent relevant study is missing in the related work and baseline methods [1].

[1] LLMs + Persona-Plug = Personalized LLMs, Liu et al., 2025.

**Questions:**

Please check my comments in Weaknesses.

---

### Official Review · Reviewer_KLps · 2025-10-31

**Soundness:** 3
**Presentation:** 4
**Contribution:** 3
**Rating:** 4
**Confidence:** 3

**Summary:**

The paper explores token-level personalization in large language models. It introduces Personal Tokens to capture user-specific expressions, a PerContrast method for estimating token-wise personalization influence through causal intervention, and a PerCE loss that re-weights tokens during training. Experiments on LongLaMP and ALOE benchmarks show consistent improvements over conventional fine-tuning methods, though gains vary across tasks.

**Strengths:**

1.The paper tackles an important yet under-explored issue: how individual tokens contribute to personalized language generation.
2.The formulation of Personal Influence Ratio (PIR) as a causal measure of token importance is clear and theoretically motivated.
3.PerCE is a simple but effective modification to standard fine-tuning, easily applicable to existing personalization pipelines.
4.Experiments across two benchmarks and multiple backbones (Qwen3, Llama3) demonstrate consistent improvements, with good reporting of stability and efficiency.
5.The presentation is generally clear.

**Weaknesses:**

1.Lack of task-completion evaluation:
Reported metrics focus almost exclusively on personalization or stylistic alignment. It remains unclear whether the proposed approach maintains factual correctness and task completion quality. Without complementary measures of accuracy or utility, the trade-off between personalization and general performance cannot be assessed.

2.Limited baseline coverage:
Comparisons are mainly against standard cross-entropy fine-tuning and simple token-identification heuristics. More competitive or recent personalization methods, such as LoRA-based adapters, prefix/prompt tuning, or preference-optimization approaches are not included.

3.Coarse analysis of PIR estimation:
The evaluation of PIR focuses on aggregate F1 scores, lacking qualitative or fine-grained visualization of how token-level importance behaves across different contexts. No analysis is provided for failure cases where neutral tokens are misclassified as personalized. Such examination would improve interpretability and help assess the robustness of PIR estimation.

**Questions:**

None

---

### Official Review · Reviewer_RvKa · 2025-10-31

**Soundness:** 3
**Presentation:** 3
**Contribution:** 2
**Rating:** 4
**Confidence:** 3

**Summary:**

This paper introduces the notion of personal tokens in the response output of LLM to capture user-specific information. It proposes a method to identify such tokens and a loss function that can increase the likelihood of such tokens occurring more in the LLM output. Experimental results suggest that if trained/finetuned with the proposed loss, the LLM produces these personal tokens more frequently.

**Strengths:**

1. The general principle proposed by the authors for personalization, which is, “focus on the output tokens, not just on what goes into the context”, seems to be a novel and necessary direction for better personalization.

2. The method proposed a method, namely PerContrast, for identifying the tokens from output that indicate personalization makes sense.

**Weaknesses:**

1. Although the research problem of leveraging/analyzing the LLM output for better personalization is important, the overall method proposed by the authors is not well justified. Why is a bag of words enough representing personal information? If yes, how? Tokens can encode persona information indirectly (through style, syntax, or topic bias), which is hard to disentangle from contextual variation.

2. The PIR is inspired by causal intervention but remains observational in practice. The paper does not verify causal consistency (e.g., through do-calculus, ablation of confounders, or alternative causal graphs). Therefore, the term “causal intervention” overstates the method’s theoretical grounding.

3. Comparison is mainly against standard CE; it would be useful to benchmark against approaches, such as, token reweighting based on attention saliency or gradient magnitude, or RLHF-style preference-based fine-tuning, etc. Lack of such baselines limits the depth of comparative analysis. Furthermore, it is not quantitatively justified, either via the experiments by authors or by citing other research, as to whether the retrieval-augmented personalization methods are not effective. So, it would make a stronger case for the acceptance of the paper if the baselines were beyond CE based training.

**Questions:**

Please see the weakness questions.

+

1. I like the causal inference based is a formulation of personal tokens. So, I am curious to know if that angel can be validated empirically. What is the underlying causal graph or how confounders (e.g., context tokens minus the user-specific prompt) are controlled when computing the counterfactual log-probabilities.

2. Could the PerCE loss potentially degrade general (non-personalized) task performance by overemphasizing personal tokens during training? It would be insightful to evaluate this by testing on a dataset, possibly synthetic or curated, where factual or context-based answers depend partially on user information but also require accurate task grounding. Such an experiment could clarify whether PerCE maintains general task fidelity while enhancing personalization.

3. What criteria were used to determine the personal tokens in the synthetic dataset?

---

### Official Review · Reviewer_Dcwm · 2025-11-03

**Soundness:** 2
**Presentation:** 3
**Contribution:** 2
**Rating:** 2
**Confidence:** 4

**Summary:**

This paper claims to focus on a novel topic in personalized LLMs, specifically on how to identify and leverage “personal tokens” that serve the user-specific needs and not the general base task. First, they introduce PerContrast that is designed to identify tokens that significantly differ in likelihood when including the persona or not with the primary query text. Thereafter, based on this difference, they propose a reweight mechanism  to train personalized LLMs that attends more focus on the identified tokens of PerContrast, suggesting these are more important for the personalization beyond the base task.

**Strengths:**

S1. This work uncovers the power of identifying and upweighting tokens that are associated with personalization in an alternating EM framework.

S2. The use of personal influence ratio in measuring the difference in PerContrast is simple, but also both interpretable and intuitive.

S3. Relatively strong performance gains; the upweighting consistently outperforms off the shelf cross entropy training and even shows transferability in personalization cross-task and cross-scenario.

**Weaknesses:**

W1. The synthetic dataset setup might be causing some bias in the comparison results especially for the personal token classification. (See related comments/questions below).

W2. It is unclear why the authors selectively picked only 3 of the 4 tasks from the LongLaMP benchmark suite for comparison. The authors mention concern of private data, but these emails were made public even before the LongLaMP dataset creation (and why it was included in that work).

W3. The experiments would have been strengthened by including other comparison methods to better understand the significance of the performance gains. Could this not be stacked on top of existing personalized LLM techniques? In the current stage the main results are lacking.

W4. Additional evaluation metrics should be included, e.g., the benchmark even also suggests METEOR in addition to ROUGE-L, but LLM-as-judge for quality is also quite popular and provides a different perspective on the quality of the generated personalized output beyond subsequence matching. If already using humans for evaluation in the keyword extraction, it would seem intuitive to also use them for some case studies on the final evaluation on the personalization task as well.

**Questions:**

1. Did you conduct any empirical analysis on how prevalent the claimed personal tokens are in real data? Is there a way to evaluate this?

2. What is the total effective compute efficiency? For example, how many alternating iterations are needed and investigation on that convergence across iterations for the proposed framework? The efficiency of “per-step” is given in Table 6, but this does not take into consideration the full iterative setup and only the inclusion of a reweight term.

3. What is the significance in explicitly conditioning the LLM-as-judge to not identify keywords from the persona? Based on the high precision of word-matching, wouldn’t it be reasonable to allow LLM-as-judge to identify such keywords from the persona? Note that this aspect was not mentioned in the main text, but appeared in the provided prompt.

4. As it is mentioned that GPT-4o is used to identify the personal tokens as ground-truths followed by human annotators, could you provide the reasoning why in your presented first example (Appendix E) that the ground truth is “deer” because to my knowledge there is nothing in the persona that would encourage personalization towards “deer” in the response “Got a deer last season.” to the query “Ever catch anything on your hunts?” This raises some concern on the quality of this generated dataset; also, 100 Q/A pairs is quite low to ensure reasonable performance comparison.

5. Please include +- in the results tables instead of only reporting the mean values.

6. Is there any theoretical justification for the design choice of PerCE as compared to any other way of leveraging the ideas gained with PerContrast (i.e., the idea/existence of personal tokens)?

7. Note that the example used throughout the paper “Where are you work at?” Is not grammatically correct, perhaps “Where do you work?”, “Where are you working?”, etc.

Please note that I am willing to consider changing my score based on your responses.

---

### Meta-Review · Area_Chair_Jnna · 2025-12-09

**Summary:**

This paper claims to focus on a novel topic in personalized LLMs, specifically on how to identify and leverage “personal tokens” that serve the user-specific needs and not the general base task.

### Pros
* This work uncovers the power of identifying and upweighting tokens that are associated with personalization in an alternating EM framework.
* The method proposed a method, namely PerContrast, for identifying the tokens from output that indicate personalization makes sense.

### Cons
* The synthetic dataset setup might be causing some bias in the comparison results
* The experimental comparisons are somewhat weak.
* Evaluation is limited to the LongLaMP dataset

### AC's evaluation

Since all reviewers vote for rejection, and no rebuttals are provided, so this will be definitely rejected.

**Reviewer Concerns:**

All concerns are outstanding, since no rebuttals are provided.

**Reviewer Scores:**

No one will increase scores, since no rebuttals are provided.

---

### Decision · Program_Chairs · 2026-01-26

Reject